# Is *Wolbachia* a sustainable component for dengue control?

**Narayanasamy Sivagnaname[1/+], Sivagnaname Yuvarajan[2], Raja Mahendran[3]**

[1]Indian Council of Medical Research, Vector Control Research Centre, Puducherry, India
[2]Sri Mankula Vinayagar Medical College and Hospital, Puducherry, India
[3]International Pest Business Consultant, Sydney, Australia

*Wolbachia*-based vector control has emerged as one of the most innovative biological interventions in contemporary dengue prevention, offering an ecologically grounded alternative to insecticide-dependent strategies. By exploiting cytoplasmic incompatibility (CI) and strain-specific viral interference, *Wolbachia pipientis*-transinfected *Aedes aegypti* (Linnaeus, 1762) mosquitoes can suppress or replace competent vector populations, thereby reducing dengue virus (DENV) transmission potential. Large-scale field deployments in Australia, Indonesia and Brazil have reported substantial reductions in dengue incidence approaching 70-80% in selected settings, demonstrating proof of principle. However, the long-term sustainability of this approach remains uncertain. Accumulating field evidence indicates that *Wolbachia* persistence and epidemiological impact are highly context-dependent, shaped by mosquito fitness, strain-host compatibility, climatic sensitivity, urban ecology and operational continuity. Experiences from Vietnam, Thailand and parts of Brazil illustrate that stable introgression cannot be assumed and may decline when releases are interrupted or ecological conditions are unfavourable. Operational constraints — including mass-rearing quality, dispersal limitations, surveillance intensity and financial costs — pose additional scalability challenges. This perspective critically evaluates whether *Wolbachia* can function as a sustainable dengue control strategy across diverse endemic settings. We argue that *Wolbachia* should not be framed as a self-sustaining or stand-alone solution, but rather as a maintenance-dependent, context-sensitive intervention whose effectiveness relies on ecological tailoring, thermotolerant strains, robust post-release surveillance and integration within adaptive integrated vector management (IVM) frameworks. When deployed judiciously alongside complementary control measures, *Wolbachia* has the potential to become an important — though not exclusive — pillar of long-term dengue control.

Key words: *Wolbachia* - dengue control - cytoplasmic incompatibility - mechanical transmission - surveillance

Dengue continues to represent a major global public health challenge, placing nearly half of the world's population at risk.[1] Its sustained expansion reflects the interaction of climate variability, rapid urbanisation, population mobility and the exceptional ecological adaptability of *Aedes aegypti* (Linnaeus, 1762).[2,3] In the absence of broadly effective antivirals, a universally protective vaccine or reliable early warning systems, vector control remains the principal strategy for dengue prevention.[3] However, conventional insecticide-based approaches are increasingly constrained by cryptic breeding habitats, outdoor resting behaviour and the rapid emergence of insecticide resistance,[4] underscoring diminishing returns of traditional control paradigms.

In this context, *Wolbachia pipientis* — an intracellular bacterium naturally absent in *Ae. aegypti* — has been proposed as an alternative biological intervention. Transinfection with *W. pipientis* can reduce arboviral replication and induce cytoplasmic incompatibility (CI), enabling population replacement with mosquitoes of reduced vector competence or, in some settings, population suppression.[5,6,7,8] Early field deployments demonstrated technical feasibility, yet outcomes have been heterogeneous across ecological and urban contexts, challenging assumptions of universal effectiveness and long-term self-sustainability.[9]

Evidence increasingly indicates that successful establishment depends on sustained infection densities, strain-host compatibility, environmental stability and intensive release strategies, compounded by the limited dispersal range of *Ae. aegypti*.[6] Operational scalability is further constrained by fitness costs, mass-rearing and sex-separation requirements, regulatory oversight and long-term financial commitments. Social acceptance and community engagement remain decisive determinants of programme continuity.[7] Variability in performance across settings highlights the vulnerability of *Wolbachia*-based strategies to ecological, climatic and programmatic disruption.

Importantly, even under optimal conditions, *Wolbachia* interventions cannot address all transmission pathways nor mitigate the inherently cyclical and multifactorial nature of dengue epidemiology. Their public health value is therefore context-dependent and contingent on integration within adaptive, evidence-driven integrated vector management (IVM) frameworks rather than deployment as a stand-alone solution.[8]

This perspective critically examines the biological assumptions, operational constraints and epidemiological uncertainties surrounding *Wolbachia*-based dengue control, questioning whether it can realistically function as a sustainable intervention or whether its promise remains bounded by structural and ecological limitations.

**doi:** 10.1590/0074-02760250268
**+ Corresponding author:** sivagnaname777@yahoo.com | ⓘ https://orcid.org/0000-0003-0817-4614

**Handling editor:** Adeilton Alves Brandão | ⓘ https://orcid.org/0000-0001-5877-607X

## Ecological and operational realities

The deployment of *Wolbachia*-infected *Ae. aegypti* has generated optimism for dengue control, yet long-term success depends on ecological nuance and operational precision. Although *Ae. aegypti* was historically considered naturally *Wolbachia*-free, recent evidence reveals low-density resident strains, including wAlbB-like variants, in wild populations.[10,11] These cryptic infections introduce complexity: interactions between resident and introduced strains may alter CI, maternal transmission fidelity and viral blocking efficiency.[12] Rigorous pre-release molecular screening and phylogenetic characterisation are therefore essential.

Transmission dynamics add further complexity. Dengue exhibits cyclical peaks every two-three years, driven by fluctuating herd immunity, serotype shifts and climate variability, particularly El Niño-Southern Oscillation patterns.[13,14] Post-release reductions may coincide with natural troughs, potentially inflating perceived impact. Only sustained multi-year entomological and epidemiological monitoring can disentangle intervention effects from background variability.

Even in *Wolbachia*-dominated populations, dengue may persist via overlooked pathways — including rare vertical transmission or mechanical transfer.[15,16] Though likely minor, these routes reinforce that *Wolbachia* cannot operate in isolation. Complementary strategies — environmental management, community participation and enhanced surveillance — remain indispensable.

Operational performance depends critically on mosquito fitness. Laboratory rearing may impose trade-offs affecting longevity, fecundity and mating competitiveness.[17,18,19] For replacement strategies to succeed, infected females must sustain high maternal transmission fidelity, while males must induce robust CI without substantial fitness penalties.[20,21,22] Field assessments of dispersal, survival and reproductive performance are thus essential to prevent ecological bottlenecks that could undermine establishment.

## Balancing suppression, replacement, and integration

*Wolbachia*-based interventions operate along two principal trajectories: mosquito population suppression and population replacement. Suppression strategies rely on the release of *W. pipientis*-infected males that induce CI when mating with uninfected females, resulting in embryonic lethality and rapid reductions in mosquito density.[5,23] Because only males are released, there is no increased risk of disease transmission. However, suppression effects are inherently transient; once releases cease, mosquito populations may rebound, particularly in high-transmission urban settings.

In contrast, population replacement strategies release both males and females, enabling *Wolbachia* to establish maternally inherited infections in wild *Ae. aegypti* populations.[18,21,24] Replacement does not primarily reduce mosquito density but instead lowers vector competence by limiting dengue virus (DENV) replication and dissemination. Once *Wolbachia* reaches high prevalence, the intervention may persist without continuous releases. This apparent durability positions replacement as transformative, although establishment depends heavily on ecological suitability, strain-host compatibility and environmental stability.

An important dimension of replacement is its influence on vertical transmission. By reducing viral replication within mosquito ovaries, *Wolbachia* may disrupt transovarial transmission that otherwise allows DENV persistence during inter-epidemic periods.[18,24,25] Nevertheless, interference varies according to strain, mosquito genetic background and climatic conditions, underscoring the need for field-validated assessments before large-scale deployment.

Mechanistically, *Wolbachia*-mediated viral blocking involves immune priming, competition for host cellular resources and disruption of viral replication complexes.[26,27,28] Although these mechanisms effectively limit positive-sense RNA viruses such as DENV, ecological and evolutionary consequences require continued monitoring. Integrating suppression and replacement-tailored to local epidemiology-offers a layered strategy: suppression for rapid outbreak control and replacement for longer-term reduction in vector competence. When embedded within adaptive IVM, these approaches enhance programme resilience and sustainability.

## The conditional success of *Wolbachia*: genetic, ecological, and epidemiological dimensions

CI forms the biological engine of *Wolbachia*-based control.[22,25] Yet CI effectiveness is not universal; it depends on compatibility between released and local mosquito genotypes.[6,29,30,31] Genetic mismatches can slow spread, weaken viral blocking or compromise persistence. Matching release strains with local populations is therefore essential for successful introgression.

Environmental factors further modulate outcomes. Temperature, humidity and rainfall profoundly influence *Wolbachia* density, mosquito fitness and DENV transmission dynamics.[12,32,33,34] Elevated temperatures may reduce *Wolbachia* density, diminish viral-blocking efficiency and lower maternal transmission fidelity.[17,35] Simultaneously, warmer conditions accelerate the extrinsic incubation period of DENV, potentially offsetting gains achieved through *Wolbachia* deployment. These interactions illustrate that impact is geographically variable and environmentally contingent.

Integrative eco-epidemiological monitoring is therefore indispensable. Surveillance must encompass mosquito density, *Wolbachia* prevalence, viral circulation, climatic trends and human case incidence.[28] Without sustained multi-year data, programmes risk attributing natural epidemiological troughs to intervention success.

These realities frame *Wolbachia* not as a universal solution but as a context-sensitive innovation. Success depends on harmonising genetic compatibility, strain robustness, field fitness and climatic resilience. Adaptive governance and evidence-driven refinement remain central to durability.

## Global perspectives on *Wolbachia*-based dengue control

Field deployments provide the strongest empirical evidence of *Wolbachia*'s potential. In northern Queensland, Australia, phased releases of the wMel strain achieved near-fixation (> 90%) and sustained prevalence for over a decade without further releases.[21] This success was supported by rigorous surveillance, community engagement and operational continuity.

In Indonesia, the cluster-randomised AWED trial in Yogyakarta demonstrated a 77% reduction in laboratory-confirmed dengue and an 86% decline in hospitalisations over two years following *Wolbachia* establishment.[6] These findings provided robust epidemiological validation under controlled trial conditions.

Brazilian deployments in Rio de Janeiro and Niterói demonstrated feasibility in large, complex urban settings.[5,28] Moderate-to-high establishment was achieved despite infrastructural and socio-economic challenges, with reductions in dengue incidence approaching 69% in some areas.

However, experiences from Vietnam and Thailand illustrate operational and ecological constraints. In Nha Trang, *Wolbachia* prevalence fluctuated seasonally, and epidemiological impact was limited.[36,37] In Thailand, hybrid suppression-replacement strategies struggled to maintain stable infection frequencies.[38] Thermal stress, reduced fitness of laboratory-reared mosquitoes and complex urban ecologies limited persistence.[12,17,33]

These comparative insights are synthesised in Table, which summarises success factors, limiting determinants and cross-cutting residual risks across diverse settings.

## Silent sparks: mechanical transmission and residual risk

While *Wolbachia* blocks biological transmission of DENV, it does not address mechanical transmission (MT). MT occurs when infectious viral particles are transferred between hosts via contaminated mouthparts during interrupted feeding or multiple probing events.[39,40,41,42] Unlike biological transmission, MT is immediate and bypasses intracellular viral replication, rendering *Wolbachia*-mediated blocking irrelevant.

The epidemiological significance of MT remains debated. Experimental evidence confirms plausibility.[39,41,43] but its contribution to sustained epidemics is likely limited compared with classical biological transmission. Nonetheless, in densely populated urban environments with high biting rates, MT could act as a sporadic ignition mechanism, potentially explaining isolated cases within *Wolbachia*-dominated populations.

Addressing residual risk requires integrated strategies. *Wolbachia*-based replacement must be complemented by measures that reduce biting pressure and overall mosquito density. Environmental management, adult suppression tools and behavioural surveillance remain critical. Recognising MT reinforces the principle that *Wolbachia* functions most effectively within a broader, adaptive IVM framework.

## Discussion and conclusion

*Wolbachia*-based interventions represent a significant paradigm shift in dengue control, transitioning from chemical suppression toward biologically mediated modification of vector competence. Field evidence

TABLE

Key global insights into *Wolbachia*-based dengue control: successes, challenges and residual risks

| Dimension | Success factors (Australia, Indonesia, Brazil) | Limiting factors (Vietnam, Thailand, others) | Residual risks (Cross-cutting) |
|---|---|---|---|
| Epidemiological impact | 10 years without outbreaks (Australia); 77% reduction in confirmed cases, 86% decline in hospitalisations (Indonesia); up to 69% incidence reduction (Brazil). | Inconsistent prevalence; weak or no measurable epidemiological signal despite releases. | Possible persistence of dengue through silent or residual transmission pathways. |
| Operational strengths | Phased releases, rigorous monitoring, strong community engagement; large-scale releases feasible (Brazil: >5 million mosquitoes). | Limited release coverage, hybrid strategies with unstable *Wolbachia* frequencies, logistical challenges. | Dependence on continuous surveillance; operational fatigue in long-term programs. |
| Strain performance | Stable wMel establishment in moderate climates, high maternal transmission fidelity. | Reduced *Wolbachia* density at high temperatures and diminished viral-blocking under heat stress. | Strain-specific vulnerabilities; need for thermally robust and high-fitness variants. |
| Mosquito fitness | Sustained maternal transmission, effective cytoplasmic incompatibility (CI) induction, stable prevalence. | Fitness costs in mass-reared mosquitoes: reduced survival, fecundity, delayed development, poor male competitiveness. | Risk of ecological bottlenecks undermining long-term establishment. |
| Community & social acceptance | High trust and participation (Australia, Indonesia); gradual acceptance in Brazil through targeted outreach. | Initial scepticism, slower adoption in areas with fragmented social trust. | Sustained engagement is essential; misinformation or distrust could reverse gains. |
| Ecological & climatic context | Moderate climates with stable *Wolbachia* persistence. | Seasonal rainfall, urban ecological complexity, and high ambient temperatures disrupt prevalence. | Climate change may destabilise *Wolbachia* density and viral-blocking capacity. |
| Integration with broader control | Complementary to existing vector control programs; strengthens integrated vector management. | Over-reliance on *Wolbachia* without supportive measures weakens outcomes. | Residual reliance on insecticides, surveillance, and community actions is inevitable. |

from Australia, Indonesia and Brazil demonstrates that, under favourable ecological and operational conditions, near-fixation of *Wolbachia* can be achieved and sustained, resulting in substantial reductions in dengue transmission.[5,6,21]

Conversely, heterogeneous outcomes in Vietnam, Thailand and complex urban contexts highlight ecological and operational fragility.[36,37,38] Thermosensitivity of key strains, reduced fitness of mass-reared mosquitoes and climatic variability may undermine persistence. Emerging data further suggest that long-term introgression may decline when operational support weakens.[9]

Residual transmission pathways — including vertical and mechanical transmission[39,40,41,42,43] and dengue's inherently cyclical epidemiology necessitate prolonged, multi-year evaluation before claims of durable elimination can be substantiated.

Collectively, current evidence indicates that *Wolbachia* is neither a universal remedy nor an inherently self-sustaining intervention. Rather, it is a powerful but conditional tool whose long-term success depends on genetic compatibility, ecological resilience, operational continuity and integration with complementary vector control strategies. When embedded within adaptive, evidence-driven IVM frameworks, *Wolbachia* can substantially reduce dengue transmission risk. Its enduring value lies not in replacing existing tools but in strengthening them — advancing dengue control toward a systems-based, sustainable prevention paradigm.

## AUTHORS' CONTRIBUTION

NS conceptualised the manuscript; SY contributed to literature synthesis and drafting; RM critically revised the manuscript. All authors approved the final version.

## DATA AVAILABILITY

No new data were generated or analysed in this study. All information is derived from previously published literature.

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

# OPEN PEER REVIEW

Memórias do IOC thanks the anonymous reviewers for their contribution to the peer review of this work.

**FIRST REVIEW ROUND**

REVIEWERS' COMMENTS

**REVIEWER #1**

The manuscript provides a critical assessment of the use of Wolbachia as a tool for the control of dengue and other arboviruses, avoiding the portrayal of this strategy as a "silver bullet" or a universally applicable solution.

This approach is fully consistent with the findings reported by Pavan et al. (2025), which demonstrate, in the context of Rio de Janeiro, that the persistence of the wMel strain may be compromised by fitness costs, vector management pressures, and environmental conditions, thereby reinforcing the need for integration with Integrated Vector Management (IVM) strategies.

Robust evidence recently presented by Pavan et al. (2025), published in PLOS Neglected Tropical Diseases, provides independent long-term data from Rio de Janeiro, Brazil, showing that the persistence of the wMel strain is fragile, reversible, and dependent on specific external conditions, and therefore cannot be considered a self-sustaining method.

In Rio de Janeiro, introgression of the wMel strain never exceeded suboptimal levels (~32%), even after massive releases (~67 million mosquitoes). Following the interruption of releases, Wolbachia frequency declined sharply, reaching <10% within less than two years, with no evidence of sustained spatial expansion. On the contrary, a contraction of the geographical distribution of infection was observed. These results directly refute the notion of autossustentabilidade (self-sustainability).

In Brazil, available data further demonstrate that the wMel strain imposes significant fitness costs on mosquitoes, resulting in a marked reduction in egg viability, impaired formation of an egg bank, and lower adult recruitment rates following population disturbances.

Although the submitted manuscript advocates for the integration of Wolbachia into vector control programs, it does not sufficiently acknowledge evidence indicating that routine vector control practices may compromise the long-term persistence of Wolbachia.

The manuscript interprets initial reductions in disease incidence as evidence of sustained effectiveness. However, the epidemiological reductions observed in Rio de Janeiro were modest (38% for dengue and 10% for chikungunya) and coincided with specific periods of chemical control and natural fluctuations in transmission.

In light of the robust evidence available in the literature, it can be concluded that the manuscript relies on conceptual premises that overlook or minimize contradictory findings. The authors overestimate the self-sustainability (which has been refuted in the literature), ecological stability, and epidemiological impact of Wolbachia, thereby posing a risk of scientific misinformation by proposing a structurally fragile intervention as a self-sustaining public health policy.

Recommendations
The manuscript would benefit from a more comprehensive discussion of the feasibility of the Wolbachia method, taking into account:

i) the very high financial costs to governments associated with large-scale mosquito releases (probably several million dollars in a small city);

ii) the lack of self-sustainability of the method, which may be linked to climatic conditions, urban ecology, and population dynamics;

iii) factors affecting the maintenance of Wolbachia, including interference with vertical transmission and reductions in bacterial density within mosquitoes;

iv) operational challenges related to implementation in highly vulnerable and high-risk areas (e.g., informal settlements or "favelas");

v) a balanced comparison between the Wolbachia approach and other preventive measures, such as dengue vaccines (e.g., the tetravalent single-dose dengue vaccine produced by the Instituto Butantan in Brazil). The authors are encouraged to critically evaluate the advantages and limitations of alternative dengue prevention strategies.

AUTHORS' RESPONSE TO THE REVIEWERS

Responses to the Reviewer's Comments
(i) High financial costs of large-scale mosquito releases
We thank the reviewer for highlighting the critical issue of financial sustainability. In response, we have revised the Discussion to explicitly address the substantial economic demands associated with large-scale Wolbachia

deployments. The revised text now details the cumulative costs of mass mosquito production, quality assurance, phased and repeated releases, intensive post-release entomological surveillance, and sustained community engagement. Drawing on documented field experiences, we acknowledge that programmatic expenditures may reach several million USD even for medium-sized urban settings, particularly where repeated releases are required to maintain introgression. By situating Wolbachia implementation within a framework of long-term financial and institutional commitment—rather than assuming biological persistence alone—the manuscript now offers a more realistic, transparent, and policy-relevant assessment of scalability.

(ii) Lack of self-sustainability under climatic, ecological, and demographic variability

We fully concur with the reviewer that emerging field evidence challenges assumptions of inherent Wolbachia self-sustainability. Accordingly, the manuscript has been revised to explicitly incorporate empirical findings from Rio de Janeiro, where wMel introgression failed to exceed suboptimal levels (approximately 32%) despite extensive releases and declined rapidly to below 10% within two years following cessation of releases, accompanied by spatial contraction, as reported by Pavan et al. (2025). We have removed any implicit claims of autonomous persistence and replaced them with an evidence-based framework of conditional persistence, dependent on ecological suitability, operational continuity, and environmental pressures. This revision ensures closer alignment with contemporary field-derived data and avoids overgeneralization.

(iii) Biological and operational determinants of Wolbachia maintenance failure

We appreciate the reviewer's request for a more detailed examination of mechanisms undermining Wolbachia persistence. In response, this section has been substantially strengthened. The revised manuscript expands discussion on temperature-driven reductions in Wolbachia density, disruptions in maternal transmission fidelity, and fitness-related dilution during population bottlenecks. We further clarify how routine vector control interventions—both chemical and environmental—may inadvertently destabilize Wolbachia maintenance by reducing adult survival, egg viability, and recruitment from egg banks. These additions emphasize that maintenance failure is not merely theoretical but biologically plausible and operationally relevant in endemic transmission settings.

(iv) Operational constraints in vulnerable urban environments

We agree with the reviewer that implementation challenges in informal and socioeconomically vulnerable urban settings require explicit attention. Accordingly, we have added a dedicated subsection addressing logistical and operational constraints such as restricted physical access, high housing density, irregular water storage practices, and fragmented governance structures. The revised text explains how these factors can reduce release efficiency and compromise surveillance sensitivity precisely in high-risk transmission environments. We further underscore the importance of sustained community engagement and adaptive, context-specific operational strategies, thereby strengthening the manuscript's relevance to real-world public health programs.

(v) Comparison with alternative preventive strategies, including dengue vaccines

We thank the reviewer for recommending a broader comparative perspective. In response, we have incorporated a new section critically comparing Wolbachia-based vector control with dengue vaccination strategies, including the single-dose tetravalent vaccine developed by Instituto Butantan. The revised manuscript discusses the relative strengths and limitations of vaccines—such as partial efficacy, serotype-dependent protection, and deployment constraints—in relation to Wolbachia-based approaches. Importantly, we emphasize that neither intervention is sufficient in isolation, reinforcing the need for integrated, layered dengue prevention strategies rather than reliance on any single technological solution.

Concluding Response

In summary, the manuscript has been substantially revised to explicitly reject assumptions of inherent self-sustainability, incorporate robust contradictory field evidence, clarify financial, ecological, and operational limitations, and reposition Wolbachia as a conditional, maintenance-dependent component of integrated vector management rather than a stand-alone solution. We sincerely thank the reviewer and the Editors for their insightful and constructive comments, which have significantly strengthened the scientific rigor, balance, and public health relevance of the manuscript.

## SECOND REVIEW ROUND

### REVIEWERS' COMMENTS

#### REVIEWER #1

No comments.

