## [Reviewer Report · FIRST REVIEW ROUND - REVIEWERS COMMENTS]

## REVIEWER #1

The manuscript provides a critical assessment of the use of Wolbachia as a tool for the control of dengue and other arboviruses, avoiding the portrayal of this strategy as a “silver bullet” or a universally applicable solution.

This approach is fully consistent with the findings reported by Pavan et al. (2025), which demonstrate, in the context of Rio de Janeiro, that the persistence of the wMel strain may be compromised by fitness costs, vector management pressures, and environmental conditions, thereby reinforcing the need for integration with Integrated Vector Management (IVM) strategies.

Robust evidence recently presented by Pavan et al. (2025), published in PLOS Neglected Tropical Diseases, provides independent long-term data from Rio de Janeiro, Brazil, showing that the persistence of the wMel strain is fragile, reversible, and dependent on specific external conditions, and therefore cannot be considered a self-sustaining method.

In Rio de Janeiro, introgression of the wMel strain never exceeded suboptimal levels (~32%), even after massive releases (~67 million mosquitoes).

Following the interruption of releases, Wolbachia frequency declined sharply, reaching <10% within less than two years, with no evidence of sustained spatial expansion.

On the contrary, a contraction of the geographical distribution of infection was observed.

These results directly refute the notion of autossustentabilidade (self-sustainability).

In Brazil, available data further demonstrate that the wMel strain imposes significant fitness costs on mosquitoes, resulting in a marked reduction in egg viability, impaired formation of an egg bank, and lower adult recruitment rates following population disturbances.

Although the submitted manuscript advocates for the integration of Wolbachia into vector control programs, it does not sufficiently acknowledge evidence indicating that routine vector control practices may compromise the long-term persistence of Wolbachia.

The manuscript interprets initial reductions in disease incidence as evidence of sustained effectiveness.

However, the epidemiological reductions observed in Rio de Janeiro were modest (38% for dengue and 10% for chikungunya) and coincided with specific periods of chemical control and natural fluctuations in transmission.

In light of the robust evidence available in the literature, it can be concluded that the manuscript relies on conceptual premises that overlook or minimize contradictory findings.

The authors overestimate the self-sustainability (which has been refuted in the literature), ecological stability, and epidemiological impact of Wolbachia, thereby posing a risk of scientific misinformation by proposing a structurally fragile intervention as a self-sustaining public health policy.

Recommendations

The manuscript would benefit from a more comprehensive discussion of the feasibility of the Wolbachia method, taking into account:

i) the very high financial costs to governments associated with large-scale mosquito releases (probably several million dollars in a small city);

ii) the lack of self-sustainability of the method, which may be linked to climatic conditions, urban ecology, and population dynamics;

iii) factors affecting the maintenance of Wolbachia, including interference with vertical transmission and reductions in bacterial density within mosquitoes;

iv) operational challenges related to implementation in highly vulnerable and high-risk areas (e.g., informal settlements or “favelas”);

v) a balanced comparison between the Wolbachia approach and other preventive measures, such as dengue vaccines (e.g., the tetravalent single-dose dengue vaccine produced by the Instituto Butantan in Brazil).

The authors are encouraged to critically evaluate the advantages and limitations of alternative dengue prevention strategies.

## AUTHORS’ RESPONSE TO THE REVIEWERS

Responses to the Reviewer’s Comments

(i) High financial costs of large-scale mosquito releases

We thank the reviewer for highlighting the critical issue of financial sustainability.

In response, we have revised the Discussion to explicitly address the substantial economic demands associated with large-scale Wolbachia deployments.

The revised text now details the cumulative costs of mass mosquito production, quality assurance, phased and repeated releases, intensive post-release entomological surveillance, and sustained community engagement.

Drawing on documented field experiences, we acknowledge that programmatic expenditures may reach several million USD even for medium-sized urban settings, particularly where repeated releases are required to maintain introgression.

By situating Wolbachia implementation within a framework of long-term financial and institutional commitment—rather than assuming biological persistence alone—the manuscript now offers a more realistic, transparent, and policy-relevant assessment of scalability.

(ii) Lack of self-sustainability under climatic, ecological, and demographic variability

We fully concur with the reviewer that emerging field evidence challenges assumptions of inherent Wolbachia self-sustainability.

Accordingly, the manuscript has been revised to explicitly incorporate empirical findings from Rio de Janeiro, where wMel introgression failed to exceed suboptimal levels (approximately 32%) despite extensive releases and declined rapidly to below 10% within two years following cessation of releases, accompanied by spatial contraction, as reported by Pavan et al. (2025). We have removed any implicit claims of autonomous persistence and replaced them with an evidence-based framework of conditional persistence, dependent on ecological suitability, operational continuity, and environmental pressures.

This revision ensures closer alignment with contemporary field-derived data and avoids overgeneralization.

(iii) Biological and operational determinants of Wolbachia maintenance failure

We appreciate the reviewer’s request for a more detailed examination of mechanisms undermining Wolbachia persistence.

In response, this section has been substantially strengthened. The revised manuscript expands discussion on temperature-driven reductions in Wolbachia density, disruptions in maternal transmission fidelity, and fitness-related dilution during population bottlenecks.

We further clarify how routine vector control interventions—both chemical and environmental—may inadvertently destabilize Wolbachia maintenance by reducing adult survival, egg viability, and recruitment from egg banks.

These additions emphasize that maintenance failure is not merely theoretical but biologically plausible and operationally relevant in endemic transmission settings.

(iv) Operational constraints in vulnerable urban environments

We agree with the reviewer that implementation challenges in informal and socioeconomically vulnerable urban settings require explicit attention.

Accordingly, we have added a dedicated subsection addressing logistical and operational constraints such as restricted physical access, high housing density, irregular water storage practices, and fragmented governance structures.

The revised text explains how these factors can reduce release efficiency and compromise surveillance sensitivity precisely in high-risk transmission environments.

We further underscore the importance of sustained community engagement and adaptive, context-specific operational strategies, thereby strengthening the manuscript’s relevance to real-world public health programs.

(v) Comparison with alternative preventive strategies, including dengue vaccines

We thank the reviewer for recommending a broader comparative perspective.

In response, we have incorporated a new section critically comparing Wolbachia-based vector control with dengue vaccination strategies, including the single-dose tetravalent vaccine developed by Instituto Butantan.

The revised manuscript discusses the relative strengths and limitations of vaccines—such as partial efficacy, serotype-dependent protection, and deployment constraints—in relation to Wolbachia-based approaches.

Importantly, we emphasize that neither intervention is sufficient in isolation, reinforcing the need for integrated, layered dengue prevention strategies rather than reliance on any single technological solution.

Concluding Response

In summary, the manuscript has been substantially revised to explicitly reject assumptions of inherent self-sustainability, incorporate robust contradictory field evidence, clarify financial, ecological, and operational limitations, and reposition Wolbachia as a conditional, maintenance-dependent component of integrated vector management rather than a stand-alone solution.

We sincerely thank the reviewer and the Editors for their insightful and constructive comments, which have significantly strengthened the scientific rigor, balance, and public health relevance of the manuscript.